# Accuracy of a New Pulse Oximetry in Detection of Arterial Oxygen Saturation and Heart Rate Measurements: The SOMBRERO Study

**DOI:** 10.3390/s22135031

**Published:** 2022-07-03

**Authors:** Stefano Marinari, Pasqualina Volpe, Marzia Simoni, Matteo Aventaggiato, Fernando De Benedetto, Stefano Nardini, Claudio M. Sanguinetti, Paolo Palange

**Affiliations:** 1Respiratory Disease Unit, G. Mazzini Hospital, 64100 Teramo, Italy; 2Presidio PneumoTisiologico Territoriale ASL 02, 66100 Chieti, Italy; pasqualina.volpe@virgilio.it; 3CNR Institute of Clinical Physiology, 56124 Pisa, Italy; marzia_simoni@libero.it; 4Independent Researcher, 73100 Lecce, Italy; matteo.aventaggiato@gmail.com; 5FISAR Foundation (Fondazione Salute Ambiente e Respiro), 66100 Chieti, Italy; debened@unich.it; 6Italian Multidisciplinary Respiratory Society, 20127 Milan, Italy; snardini.pneumologo@gmail.com; 7Editorial Activities, Italian Multidisciplinary Respiratory Society, 20127 Milan, Italy; sanguinetticlaudiomaria@gmail.com; 8Department of Public Health and Infectious Diseases, Division of Respiratory Diseases, Umberto I Hospital, Sapienza University, 00161 Rome, Italy; paolo.palange@uniroma1.it

**Keywords:** wrist-watch pulsoximeter, chronic cardiac diseases, chronic respiratory diseases, heart rate, pulse oxygen saturation, reflective pulseoximeter, SpO_2_

## Abstract

Early diagnosis and continuous monitoring of respiratory failure (RF) in the course of the most prevalent chronic cardio-vascular (CVD) and respiratory diseases (CRD) are a clinical, unresolved problem because wearable, non-invasive, and user-friendly medical devices, which could grant reliable measures of the oxygen saturation (SpO_2_) and heart rate (HR) in real-life during daily activities are still lacking. In this study, we investigated the agreement between a new medical wrist-worn device (BrOxy M) and a reference, medical pulseoximeter (Nellcor PM 1000N). Twelve healthy volunteers (aged 20–51 years, 84% males, 33% with black skin, obtaining, during the controlled hypoxia test, the simultaneous registration of 219 data pairs, homogeneously deployed in the levels of Sat.O_2_ 97%, 92%, 87%, 82% [ISO 80601-2-61:2017 standard (paragraph EE.3)]) were included. The paired T test 0 and the Bland-Altman plot were performed to assess bias and accuracy. SpO_2_ and HR readings by the two devices resulted significantly correlated (r = 0.91 and 0.96, *p* < 0.001, respectively). Analyses excluded the presence of proportional bias. For SpO_2_, the mean bias was −0.18% and the accuracy (A_RMS_) was 2.7%. For HR the mean bias was 0.25 bpm and the A_RMS_3.7 bpm. The sensitivity to detect SpO_2_ ≤ 94% was 94.4%. The agreement between BrOxy M and the reference pulse oximeter was “substantial” (for SpO_2_ cut-off 94% and 90%, k = 0.79 and k = 0.80, respectively). We conclude that BrOxy M demonstrated accuracy, reliability and consistency in measuring SpO_2_ and HR, being fully comparable with a reference medical pulseoxymeter, with no adverse effects. As a wearable device, Broxy M can measure continually SpO_2_ and HR in everyday life, helping in detecting and following up CVD and CRD subjects.

## 1. Introduction

Chronic obstructive pulmonary disease (COPD) and Heart failure (HF) are non-communicable diseases (NCDs), responsible for a huge number of deaths and years of life lost (YLL) [1,2,3,4,5,6,7,8,9,10]. Initial symptoms of COPD and HF can be elusive and easily confused with normal ageing. When both conditions exacerbate, hospitalization and even death can follow [11,12,13,14,15,16,17,18,19,20]. Many COPD subjects in the last stages of their disease need long-term continuous oxygen therapy (LTOT). Similarly, diagnosis and monitoring can be difficult in other prevalent respiratory chronic conditions (Idiopathic Pulmonary Fibrosis (IPF) and other interstitial lung diseases (ILD)) [21,22] and Obstructive Sleep Apnea Syndrome (OSAS) [23,24] since symptoms of all these chronic diseases can vary significantly daily. Non-invasive monitoring of vital parameters through reliable devices would be welcome in providing a careful surveillance of subjects [25,26,27,28] as well as in identifying early the disease or the need of LTOT, or the persistence of its indication over time [29]. A clinical-functional monitoring in real time of normal daily activities and night rest can help in defining the causes of symptoms [30,31] and in identifying sleep disordered breathing [32]. Summarizing, chronic heart and pulmonary conditions need to be identified as early as possible and, after diagnosis, to be monitored as easily as possible, better if in real life. Blood oxygen saturation (SpO_2_) and heart rate (HR) are universally kept as a reliable proxy for heart and lung performance.

For these reasons, devices and technologies are being studied, mainly in the field of tele-monitoring but also in designing new tools to meet unmet needs. One line of research is the continuing ambulatory recording of vital signs such as SpO_2_ and HR in real life to acquire information for early diagnosis, treatment and its effectiveness. The concept “historical predecessors” are the Holter Monitor and the device for ambulatory blood pressure monitoring.

In the last years, many devices have been produced to assist people in their leisure activities. So-called smartwatches are able to measure heart frequency and correlate it with movements. A smaller number of devices has been marketed with improved functions, e.g., measuring not only HR but also SpO_2_. However, these commercial devices for fitness are not compliant with a specific set of regulations requested for the medical devices, and therefore, they cannot be adopted for clinical uses. There are only a few medical devices certified for clinical uses (patients’ monitoring, follow-up, early-diagnosis, and professional healthcare treatments) [33,34,35,36,37]. Most of these devices are invasive, wire, or they require hospitalization for monitoring.

Devices wrist-wereable are based on reflective photoplethysmography, which is particularly complex compared to the classic transmission technology. Based on our knowledge, there is at least one study which reported the ability of reflective photoplethysmography in measuring SpO_2_ levels with adequate accuracy [38], but on our clinical experience, measuring only SpO_2_ is not useful enough in evaluating patient’s situation. A more precise judgement is provided by the simultaneous measurement of HR, since cardiac arrhythmias can interfere with the right interpretation of SpO_2_ data. Furthermore, the simultaneous measurements of SpO_2_ and HR can help diagnose other pathological conditions such as sleep disordered breathing [39,40].

The continuous monitoring in real daily life can give important information for diagnosis/management of cardio-respiratory diseases.

For this purpose, a new pulse oximeter was designed, wearable such as a watch and able to correlate pulse and oximetry with exercise. This new device, named BrOxy M (CE marked, patented: WO 2019/193196 A1; WO 2021/069729 A1), aimed to fill the gap of currently available devices, with a high-accuracy level of measurements, due to a unique calibration system “tailored on patients”.

The aims of the present study of controlled desaturation were: (i) to confirm the performance of the BrOxy M in comparison with a reference, CE marked, pulse oximeter equipment widely used in clinical settings; (ii) to assess possible factors affecting BrOxy M performance.

## 2. Materials and Methods

### 2.1. Study Design

This monocentric interventional post-market clinical study (Annex I of EN ISO 14155:2020, www.iso.org) (accessed on 29 June 2021), denominated SOMBRERO (Italian acronym for “SOrveglianza delle Misure di BroxyM durante REspirazione di diffeRenti concentrazioni di Ossigeno”- i.e., “Validation of measures taken by Broxy during respiration of different oxygen concentrations”), was performed in Italy according to the procedures described in the standard ISO 80601-2-61:2017 “Medical electrical equipment—Part 2-61: particular requirements for basic safety and essential performance of pulse oximeter equipment”. The study, comparing SpO_2_ and HR recorded with the BrOxy M pulse oximeter and with a reference pulse oximeter equipment, was performed in healthy volunteers in controlled desaturation conditions over a range between 80% and 100% SpO_2_. It was chosen to consider the reference range 80% to 100% since this range includes normal values (≥94%) and dangerous values in the range 80–90%, which are most frequently observed in domiciliary monitoring or outpatient clinics. Values under 80% need immediately a monitoring in a hospital setting.

The study was approved by the Chieti University Ethical Committee (district of Chieti and Pescara during the session no. 6 of 11 March 2021- Protocol number: 01M2020-CH.LMD).

All participants gave their written informed consent.

### 2.2. Materials

BrOxy M is a Class IIa, CE-marked, medical wearable device, with a size of a wristwatch. It consists of a bracelet with an adjustable strap that allows the device to be adapted to the subject’s wrist (Figure 1), a charging base (Figure 2), and a medical software for Personal Computer. The sensor that detects photoplethysmographic signals is on the inner face of the device (see Figure 1). It is suggested to apply this sensor about 3 or 4 cm from the wrist, close to the ulna, preferably on the portion of skin that covers a visible blood vessel. The device is then applied to the subject’s wrist by fastening the strap so that the fastening is firm, but not excessively tight. The adhesion between the skin and the device is ensured by a single-use, medical grade, double-sided adhesive film.

Thanks to the sensor (mod. MAX 30102, produced by Maxim Integrated^®^) placed on the side in contact with the skin of the subject’s wrist, BrOxy M is able to acquire photoplethysmographic signals in the frequency of red (660 nm) and infrared (880 nm) light and memorize them, with a sampling frequency of 50 Hz and a resolution of 18 bits. According to the product intended use, registration of photoplethysmographic signals using BrOxy M can occur for up to 24 h having a 16 Mbytes memory and a Li-Ion battery with a capacity of 900 mAh. In this study, signal registration occurred for a maximum of around 35 min for each enrolled subject. At the end of the signal registration phase, BrOxy M will be placed on the one-station charging base. Recorded signals were automatically transferred to the PC in use for the study and archived in an electronic medical record set up for each enrolled subject. The signals recorded by the device were then processed by a Sponsor delegate with the software for medical use provided by Life Meter S.r.l.

The raw signals recorded have been analyzed on the PC by the BrOxy M software that integrates an algorithm that operates according to the following processing steps:
From the signals recorded with the triaxial accelerometric sensor, the jolt (i.e., the derivative in time of acceleration) is calculated as shown in the following equation:
$$j\left[t\right]=\sqrt{x{\left[t\right]}^{2}+y{\left[t\right]}^{2}+z{\left[t\right]}^{2}}-\sqrt{x{\left[t-1\right]}^{2}+y{\left[t-1\right]}^{2}+z{\left[t-1\right]}^{2}}$$ where *j*[*t*] is the instantaneous jolt value at the time *t* and *x*, *y*, and *z* are the instantaneous values of the tri-axial acceleration (arbitrary units) detected from the accelerometer sensor at time *t* and *t* − 1.The time intervals relating to an absence of movement are selected by applying an experimentally determined threshold (threshold value = 18 arbitrary units) on the absolute value of jolt.By selecting the red and infrared (IR) signals within the time intervals related to the quiet state of the subject, appropriate bandpass filtering (digital ant causal finite impulse response (FIR) filter of 50th order, bandwidth from 0.5 Hz to 3 Hz) is applied to extract a relevant section of signal that allows to calculate your heart rate.From the same raw signals, the following components are extracted:
*RED_DC* = continuous part of the red signal (mean value calculated from the epoch of red signal);*RED_AC* = alternating part of the red signal (root mean square value from the epoch of red signal filtered as in point 3);*IR_DC* = continuous part of the infrared signal (mean value from the epoch of *IR* signal);*IR_AC* = alternating part of the infrared signal (root mean square value calculated from the epoch of *IR* signal as in point 3);
From these parameters we calculate the value of the gamma parameter with the equation shown below:
γ=ln(IR_ACIR_DC)ln(RED_ACRED_DC)From the value of γ we calculate the SpO_2_ using the following equation:$${\mathrm{SpO}}_{2}=a\cdot \gamma^{2}+b\cdot \gamma +c$$
where *a*, *b* and *c* are experimentally calculated and constant parameters for each subject analyzed (*a* = −44.6, *b* = 5.9; *c* = 108.1).


Where *a*, *b* and *c* are experimentally calculated and constant parameters for each subject analyzed.

Each SpO_2_ measurement is corrected by means of the intra-subject calibration procedure described in the patents WO2019/193196 A1 and WO2021/069729 A1 (see below, paragraph 2.5, point b). This technical procedure allows for every subject to extract after a 90-s finger recording an average value of ${{\mathrm{SpO}}_{2}}^{\mathrm{finger}})$, which is needed to calculate a specific value—offset—between finger and wrist measurement. Immediately afterwards, the BrOxy M device is placed on the wrist of the subject (as described above) and signal recording is started, keeping the subject at rest condition. SpO_2_ measurements for the first 90 s of wrist recordings allow the calculation of the wrist reference SpO_2_ value (${{\mathrm{SpO}}_{2}}^{\mathrm{wrist}}$) which is used to calculate the intra-subject calibration offset as:$${{\mathrm{SpO}}_{2}}^{\mathrm{offset}}={{\mathrm{SpO}}_{2}}^{\mathrm{finger}}-{{\mathrm{SpO}}_{2}}^{\mathrm{wrist}}$$

Finally, the following correction applies to each SpO_2_ measure obtained from wrist signals:$${{\mathrm{SpO}}_{2}}^{\mathrm{compensated}}={\mathrm{SpO}}_{2}+{{\mathrm{SpO}}_{2}}^{\mathrm{offset}}$$

The used reference device was the pulse oximeter Nellcor™ Bedside Respiratory Subject Monitoring System model PM1000N (Covidien LLC, Minneapolis, MN, USA). It is a CE marked device routinely used in the hospital environment for monitoring subjects while they are in bed or performing a walking test. It is traceable to co-oximeter SaO_2_ values. The Nellcor^TM^ requires the use of the Nellcor™ Adult XL SpO_2_ Sensormod. MAXALI (sterile, single use only), that allows the storage of the displayed measurements of HR and SpO_2_, and their downloading on a PC memory via USB. 

### 2.3. Participants

Inclusion criteria for participants were being between the age of 18 and 50 years and having: (1) healthy status with no evidence of any medical problem (ASA Physical Status Classification I); (2) positive new Allen’s test; (3) intact and healthy skin on the selected wrist; (4) wrist circumference between 15 and 20 cm; (5) normal ECG; (6) the ability to understand/execute the required study procedures, as well as to write an informed consent to the study.

Participants were excluded if they had: (1) altered hemoglobin parameters (αHb ≤ 10 gr/dl or COHb ≤ 3%or MetHb ≤ 2%); (2) any cardiovascular or pulmonary pathology in medical history; (3) any episodes of respiratory infection in the past 30 days; (4) any experience of dyspnea hospitalization in the past 2 months; (5) comorbid condition(s) thought to affect adversely the participation in the study (e.g., cardiac, neurological, musculoskeletal or psychological impairments), including allergy to adhesive tapes; (6) chronic drug intake known to interfere with gaseous exchanges or to induce changes in cardiac frequency; (7) body mass index <18 kg·m^2^; (8) participation in interventional studies with a medicinal product or a medical device in the past 30 days. Women were excluded if they were pregnant.

It was estimated that a minimum target of 12 subjects, resulting in at least 200 data pairs, was sufficient to evaluate SpO_2_ accuracy with 95% of statistical power at 0.05% significance level.

### 2.4. Definition of Valid Measurement

A “valid measurement” is defined as a measurement obtained under the following conditions:Signal compliant with quality control for both the reference and BrOxy M;No movement above the predefined threshold for both the reference and BrOxy M.

The quality control of the signal is carried out automatically by the HR and SpO_2_ calculation algorithm and is based on the following conditions:
Starting from the signals, through appropriate band-pass filters, the following components are calculated as follows:RED_DC = continuous part of the red signal;RED_AC = alternating part of the red signal;IR_DC = continuous part of the infrared signal;IR_AC = alternating part of the infrared signal;
Average values of RED_DC and IR_DC have to be in the expected range, determined empirically;If the previous point occurs, peak-peak amplitude values of RED_AC and IR_AC have to be in the empirically determined acceptability range;If the previous condition is verified, the correlation coefficient calculated between IR_AC and RED_AC has to be higher than the empirically determined threshold.

The algorithm used to recognize the time windows to be excluded for motion artifacts is organized with the following steps:
Control of the amplitude of photoplethysmographic signals: the amplitude of the alternating component of red and infrared signals (obtained by filtering signals with an anti-causal high-pass filter must not exceed a threshold determined empirically);Control of the correlation between photoplethysmographic signals and accelerometric signals: the module of accelerometric signals recorded on three orthogonal axes of space (*A_x_*, *A_y_* and *A_z_*) is calculated as $\left|A_{XYZ}\left[t\right]\right|=\sqrt{A_{x}{\left[t\right]}^{2}+A_{y}{\left[t\right]}^{2}+A_{z}{\left[t\right]}^{2}}$ and, subsequently, it occurs that:
The correlation between $\left|A_{XYZ}\left[t\right]\right|$ and (photoplethysmographic) signal recorded in the red frequency is less than a certain threshold derived empirically;The correlation between $\left|A_{XYZ}\left[t\right]\right|$ and (photoplethysmographic) signal recorded in the infrared frequency is less than a certain threshold derived empirically.
If the conditions in point 1 and 2 occur, the time window from which to derive heart rate and SpO_2_ data is excluded from the collection of data deemed useful in the context of the study.


At this point, verified all the above conditions, the values of SpO_2_ and HR are calculated for the signal windows considered. These values are considered acceptable if the following conditions occur at the same time:Heart rate values between 40 and 180 bpmSpO_2_ values between 80 and 100%

### 2.5. Study Protocol and Sample

Each subject undergoing the test respected the following procedure:(a)Sitting with the right arm leant on the table, breathing room air for 30 min(b)Calibration of sensor: this procedure requires the distal phalanx of a patient’s finger (e.g., index finger) to be placed on the photoplethysmographic sensor on the BrOxy M wearable device in order to record 90 s of red and infrared signal. Using the recorded signals, the calibration algorithm is applied as described in the patents WO2019/193196 and WO2021/069729 (see Appendix A).(c)BrOxy M positioning (see Section 2.2);(d)Positioning of the single use sensor MAXALI of the reference device on the fingertip of index finger of the right hand warning the subject not to move arm and hand(e)Positioning of a single use nose clip to block the nasal airflow and start of test with the following experimental procedure: mouthpiece with sterile filter connected to a Hans Rudolph valve (one way air valve), whose inspiratory way, through a Douglas bag tubing 1.5 m long, is connected in sequence—according to time set afterwards reported, through a single channel tubing valve, to 4 (four) Douglas bag 100 lt each, continuously supplied by 4 cylinders each containing a different O_2_ concentration (see Table 1 after the following paragraph).

The aim of the test was to obtain 4 different plateaus.

For each subject, four different saturation plateaus have been set up (i.e., at 97%, 92%,87% and 82% SpO_2_), waiting for 30″ before starting to keep the SpO_2_ level on each plateau for 2.5′ in order to obtain in post-processing 8 couples of instantaneous values of SpO_2_ measured from BrOxy M and from the reference device. These measurements have been at least 20″ away from each other, for a total of 32 measurements for each subject, as summarized in the following Table 1 and diagram (Figure 3).

From the records carried out on the 12 subjects enrolled in the clinical protocol, 423 data pairs were obtained. Since one of the aims of this work is to quantify the accuracy of estimation of the SpO_2_ obtained by BrOxy M with reference to the SpO_2_ measured by the reference device, the outliers were removed by discarding data pairs that provided an error that differed from the error average by more than twice the standard deviation. Afterwards, the hypothesis that the errors came from a standard normal distribution was confirmed by means of a Kolmogorov-Smirnov test (5% level of significance, *p* = 0.059).

With regard to the efficacy of the device on subjects with dark pigmentation of the skin, in order to be able to meet FDA guidelines for pulse oximeters [41], as subsequently detailed and illustrated in the results, at least 2 darkly pigmented subjects (Fitzpatrick scale, Type V–VI) were included and more than 15% of the measurements were carried out on them.

### 2.6. Statistical Analysis

Statistical analysis was performed with SPSS, Statistical Package for the Social Sciences (SPSS Inc., Chicago, IL, USA), following the standard ISO 80601-2-61:2017 (EE.3. Procedure for non-invasive laboratory testing on healthy adult volunteers). The accuracy is stated in terms of the root-mean-square (A_RMS_) difference between the readings of the BrOxy M and the standard readings of the Nellcor^TM^, as follows:$$A_{\mathrm{RMS}}=\sqrt{\frac{\sum_{\left(n-1\right)}^{n}{\left({\mathrm{SpO}}_{2\mathrm{BrOxy}}-{\mathrm{SpO}}_{2\mathrm{Nellcor}}\right)}^{2}}{n}}$$
which corresponds to the standard deviation (SD) of the mean difference.

Mean error (bias) was quantified as mean (BrOxy M SpO_2_minus Nellcor SpO_2_).

SpO_2_ is considered *normal* (when measured at sea level) for values ≥95%, *mild hypoxemia* for values between 94% and 91%, *moderate hypoxemia* for values between 90% and 86%, and *severe hypoxemia* for values <85%. Considering that SpO_2_ < 94% has been associated with pathophysiological and clinical consequences (these latter become increasingly more serious and life-threatening with lower values), analyses were performed also in two subgroups of standard oxygen saturation: SpO_2_ ≤ 94%, and SpO_2_ ≤ 90%.

As regards HR, the mean absolute percent error (MAPE = average of the absolute percent difference between HR BrOxy and HR Nellcor/HR Nellcor) was also calculated to provide a gauge of measurement error of BrOxy M.

The comparison between BrOxy M and Nellcor^TM^ was evaluated by using (1) the Pearson correlation coefficient and the concordance correlation coefficient (perfect positive agreement at 1); (2) the paired t test 0 to determine whether the mean difference between the two instruments is zero (presence/absence of proportional bias); (3) the Bland-Altman analysis to evaluate the agreement between experimental and standard readings. [42].

If the two methods are comparable, then differences should be small, with the mean of the differences (bias) close to 0; (4) multiple regression analysis to evaluate possible significant associations of the difference with sex, age, BMI, and skin pigmentation.

By basing on two desaturation cut-off (94% and 90%), sensitivity, specificity, positive/negative predictive values were calculated, and Cohen’s Kappa coefficient was used to evaluate the agreement between the two pulse oximeters (agreement: 0.01–0.20 slight, 0.21–0.40 fair, 0.41–0.60, moderate, 0.61–0.80 substantial, 0.81–1.00 almost perfect or perfect). The significance level, that is the probability of rejecting the null hypothesis when it is true, will be set to 0.05 (two-sided, corresponding 5% risk of concluding that a difference exists when there is no actual difference).

## 3. Results

A set of 399 data pairs was considered: 73 data pairs between 80% and 86%, 117 between 87% and 93%, and 209 between 94% and 100%. In order to equalize the distribution on the three ranges, some data pairs related to the ranges between 87% and 93% and between 94% and 100% have been excluded in the following ways: for each one of these two ranges, cyclically, data pairs related to one subject at a time were considered, starting from the subject number 1 up to the subject number 12, and then starting again; for each subject’s data pairs, the last acquired data pair was removed. The removal of data pairs was carried out cyclically until the number of data pairs in the range considered was less than or equal to 73 (number of ranges between 80% and 86%) Eventually data pairs for analyses were 219, 84% measured in men (72% with white skin and 28% with black skin), and 30% in people with black skin [43]. The ISO 80601-2-61:2017 standard, in case of “Procedure for non-invasive laboratory testing on healthy volunteers” (paragraph EE.3), considers the target of 200 sets of data pairs to be statistically significant and satisfactory. Table 2 shows the sample characteristics. The participants in the study were 10 men and 2 women aged 20–51 years, including 8 with white skin and 4 with black skin [43].

Mean SpO_2_ was 90.8% (SD 6.1%), as measured with Nellcor^TM^, and it was 91.0% (SD 6.3%), as measured with BrOxy M; 67% and 47% of data pairs included standard values of SpO_2_ ≤ 94% and ≤90%, respectively. HR measured with Nellcor^TM^ and BrOxy M ranged from 64 to 122 bpm and 62 to 126 bpm, respectively.

### 3.1. Differences between the Two Pulse Oximeters

SpO_2_ readings by the two oximetry devices resulted highly positively and significantly (*p* < 0.001) correlated (Figure 4a), with Pearson correlation coefficient 0.91. High correlation was confirmed by the Lin’s concordance correlation coefficient (CCC 0.91, 95% Confidence Interval, CI, 0.88–0.93). As concerns HR (Figure 4b), the Pearson correlation coefficient was 0.96 (*p* < 0.001), and the Lin’s CCC was 0.95 (95% CI 0.94–0.96).

The correlation analysis quantifies the degree to which two variables are related, but a high correlation does not automatically imply that there is good agreement between the two devices. The paired T Test 0 provided the mean difference between BrOxy M and Nellcor^TM^, the standard deviation and the standard error of the difference, along 95% confidence interval for the mean, and the significance level for the test that the mean of the difference equals 0 (Table 3).

The more the average difference is close to 0, the greater the agreement between the two oximetry devices. In the reference range 80% to 100%, the bias resulted very close to 0 (0.18%), indicating that, on average, the BrOxy M measures 0.18 units more than Nellcor^TM^. The accuracy was 2.7% (corresponding to the standard deviation of the mean difference). The paired T test 0 confirmed the hypothesis that the average difference was not different from 0 (*p* = 0.31).

The mean bias for the subjects with Nellcor SpO_2_ ≤ 94% was 0.57%, with accuracy 3.0%, whereas corresponding figures for the subjects with Nellcor SpO_2_ ≤ 90% were 0.94% and 3.3%, respectively. It is reported in literature that pulse oximeters perform better at the higher saturation levels compared to the lower end [44]. However, there are no acceptance criteria associated with different levels of hypoxia, and when presenting the A_RMS_, the common methodology is to provide the data across the whole range (i.e., 80% to 100%, in this study).. The standard ISO 80601-2-61:2017 (Annex AA) reports that, based on clinical experience, SpO_2_ accuracy ≤4% is acceptable for many monitoring applications. Corresponding results regarding HR are shown in Table 4. The paired T test 0 confirmed the hypothesis that the average difference was not different from 0 (*p* = 0.32). HR MAPE ≤10% is acceptable [45].

Multiple regression analyses indicated no significant association of SpO_2_ difference between the two pulse oximeters with sex, age, skin color, and BMI. The results did not change after selection of data pairs with reference SpO_2_ ≤ 94% or ≤ 90% (Table 5).

SpO_2_ difference was not correlated to HR difference neither in the whole sample (Pearson r = −0.10, *p* = 0.13), nor after selecting cases with Nellcor SpO_2_ ≤ 95% (r = −0.07, *p* = 0.39) or ≤90% (r = 0.04, *p* = 0.68).

### 3.2. Bland-Altman Plot

In Figure 5, the Bland-Altman plot shows the average values of simultaneous BrOxy M SpO_2_ and Nellcor SpO_2_(*X*-axis) versus their differences (*Y*-axis). The plot quantifies the mean bias (0.18%) and the limits of agreement (−5.1% to +5.5%), within which 95% of the differences between the two instruments are included. Figure 6 shows the Bland-Altman plot for the data pair with Nellcor SpO_2_ ≤ 94% (a) and ≤90% (b).

In the whole range of Nellcor SpO_2_ (from 80% to 100%) (Figure 5), a difference less than −4% (underestimation of BrOxy M) was observed in seven cases (3.2%): in five cases the standard SpO_2_ was ≤90%, thus making the underestimation irrelevant from a clinical point of view; the same for the remaining two cases, with Nellcor SpO_2_ 100% and BrOxy M SpO_2_ 95%. The number of cases in which the standard concentration was overestimated was higher (difference > 4%, n = 21, 9.6%): the overestimation regarded 15 subjects with both Nellcor SpO_2_ and BrOxy M SpO_2_ < 90% (difference clinically irrelevant), three subjects with Nellcor SpO_2_ ≤ 90% and BrOxy M SpO_2_ in the range 91–96%, and three subjects with Nellcor SpO_2_ between 91 and 92% and BrOxy M SpO_2_ between 96 and 97%.

In Figure 7, the Bland-Altman plot shows the average values of simultaneous BrOxy M HR and Nellcor HR (*X*-axis) versus their differences (*Y*-axis). The plot quantifies the mean bias (0.25 bpm) and the limits of agreement (−7.1 bmp to +7.7 bpm), within which 95% of the differences between the two instruments are included.

### 3.3. Sensitivity, Specificity, Positive and Negative Predictive Values

Sensitivity for the ability of BrOxy M to detect Nellcor SpO_2_ ≤ 94% was 94.4%; specificity to detect normal saturation (SpO_2_ > 94%) was 83.1%; positive and negative predictive values were 91.2% and 88.9%, respectively. Sensitivity for the ability of BrOxy M to detect Nellcor SpO_2_ ≤ 90% was 86.9%; specificity to detect SpO_2_ > 90% was 92.5%; positive and negative predictive value were 90.5% and 89.5%, respectively. The agreement between BrOxy M SpO_2_ and Nellcor SpO_2_, as expressed by the Cohen’s kappa coefficient, was substantial for both the cut-off 94% (k = 0.79) and the cut-off 90% (k = 0.80), and very close to almost perfect agreement (>0.80).

## 4. Discussion

In this study we evaluated the accuracy of BrOxy M in predicting oxygen saturation and heart rate as measured by a reference device. The readings by BrOxy M were highly correlated with the simultaneous measurements obtained with the Nellcor^TM,^ a device routinely used in clinical practice all over the world. The analyses indicated no presence of proportional bias. The average difference of SpO_2_ (bias) was very close to zero (0.18%), thus indicating good agreement between the two devices; the accuracy (2.7%) was acceptable, according to the standard ISO 80601-2-61:2017. In 12.8% of cases, SpO_2_ difference was greater than 4%, both positive and negative. However, only in 6/219 cases (2.7%) under or overestimation resulted clinically relevant. The analyses indicated similar results as regards HR. In earlier studies, sex was found to be a significant predictor for bias, and dark skin pigmentation resulted in overestimation of arterial oxygen saturation especially at a low saturation in some pulse oximeters [46,47,48,49]. In our study, sex and skin color, as well as age, and BMI, did not influence the difference between the two devices.

A relevant result was that substantial agreement (very close to “almost perfect”) was found between BrOxy M SpO_2_ and Nellcor SpO_2_ when both the cut-offs 94% and 90% were considered. Finally, no adverse events related to the use of the device BrOxy M were observed, besides a temporary slight skin irritation of the wrist.

Which could be the use of a wearable and reliable pulse-oximeter able to record continually signals in real life?

Since many years we have known that monitoring subjects with pulmonary and cardiac diseases at home can decrease hospital admissions, emergency department visits and hospital length-of-stay [50,51,52,53,54,55], while increasing the patients’ quality of life [56]. A significant reduction of costs from telemonitoring in comparison to usual care was shown in heart failure [57,58]. The same advantages have been demonstrated in COPD subjects [26] with a technology easily accepted by subjects, even in decreasing emergency-room visits [28]. In cardiac and respiratory chronic diseases, early diagnosis can be challenging, and acute episodes of exacerbation can more or less rapidly modify the health status of subjects. The availability of a continuous monitoring of the vital parameters, both during the normal daily activities and the sleep period, could catch the first signs of a chronic disease.

The results of the present study indicated the BrOxy M performs such as another well-established device in use for a long time. It is user-friendly, fully wearable, and not hampering normal daily activities. Furthermore, it can record both SpO_2_ and HR.

Given its peculiar structure, BrOxy M can be worn without any trouble by the subject at any time of day or night. Using the specific software, it is possible to download the signals stored in the BrOxy M device, view, archive, and export them. These signals can be used as inputs for algorithms with diagnostic purposes to describe the physiological state of the subject.

Although the algorithms integrated in the BrOxy M software apply a particular automatic calibration procedure of the SpO_2_ estimation parameters, as described in patents WO 2019/193196 A1 and WO 2021/069729 A1, in order to overcome the difficulties introduced by reflection photoplethysmography, a feature to be emphasized is that BrOxy M does not use any AI procedures to give the outputs, such as neural network or any kinds of linear/non-linear regressors. Other devices (such as Withings ScanWatch, wrist reflection pulse oximeter) [38] use neural networks to process the signals to provide the SpO_2_ estimate by means of a “black box” approach. Similar to in other fields of medicine, the use of AI, while suitable for the majority of cases, can lead in some cases, lying out of the population data used to train the machine, to untoward results [59]. In fact, FDA requires a dedicated regulatory pathway with a careful evaluation of the risks linked to the use of devices using neural networks and other algorithms based on AI [60]. 

### Limitations

A potential limitation of our study could be that accuracy of BrOxy M was evaluated, in comparison to another pulse oximetry, which represented the gold standard, without performing arterial blood gas. The Nellcor^TM^ is a CE marked device normally used in the hospital environment, and it is traceable to co-oximeter SaO_2_ values. The acceptable agreement demonstrated between the BrOxy M and Nellcor^TM^ should guarantee adequate reliability of the results. It should be pointed out that the comparison with the arterial oxygen saturation (SaO_2_) calculated, and not directly measured in an arterial blood gases sample, may not represent the right choice since arterial oxygen partial pressure (PaO_2_), fluctuates during the respiratory phases. The present study was conducted in a controlled environment with a well-established protocol and methodology to collect SpO_2_ and HR measurements during specific plateaus of SpO_2_ in healthy participants. Its design may limit the generalizability of the results to real-world situations. In future works, the accuracy of BrOxy M should be tested in real-life conditions (i.e., at home, in hospital, and during rehabilitation), on specific populations such as subjects with COPD or obstructive sleep apnea, or to diagnose and monitor subjects with cardiac and respiratory diseases.

## 5. Conclusions

Our study, aimed at evaluating the performance of a new device in comparison with a reference, CE marked, medical pulse-oximeter, showed an acceptable accuracy (according to the standard ISO 80601-2-61:2017) for both SpO_2_ and HR, independently of gender, age, skin color, and BMI. Furthermore, the agreement between BrOxy M and Nellcor^TM^, as concerns SpO_2_ cut-off equal to 94% or 90%, resulted very close to be almost perfect. No proportional biases were detected. No relevant adverse effects occurred.

Our study is important because a device such as BrOxy M able to monitor finely and reliably subjects’ vital parameters at rest and during exercise at home, in real life, continuously for 24 h can fulfill unanswered needs in the management of chronic cardiac and respiratory conditions.

## Figures and Tables

**Figure 1 sensors-22-05031-f001:**
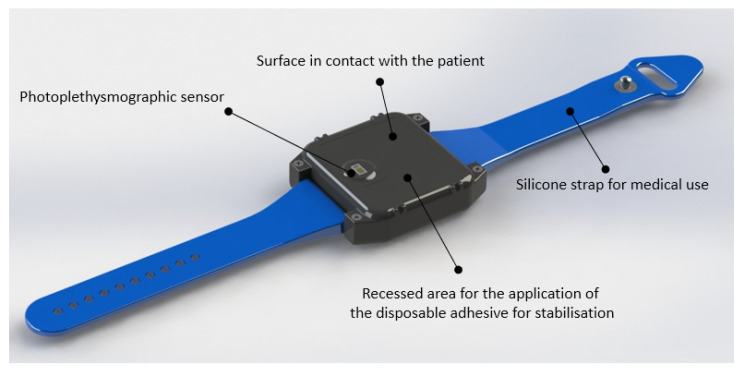
View of the wearable BrOxy M. The cover of the device is in contact with the subject and integrates the photoplethysmographic sensor and the recessed area for the application of the disposable adhesive (medical grade) used to stabilize the fixing of the sensor.

**Figure 2 sensors-22-05031-f002:**
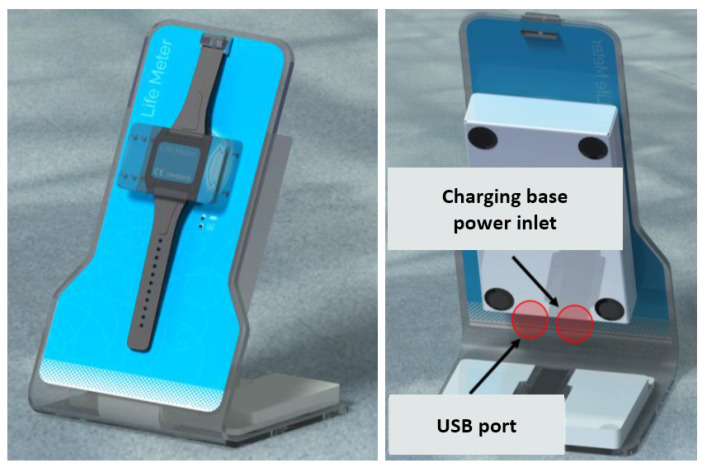
Front (**left**) and rear (**right**) view of the BrOxy M device charging base. The charging base is used to carry out wireless charging of the Li-Ion battery integrated into the BrOxy M device. Moreover, the charging base connects via USB to the PC and via Wi-Fi to the BrOxy M device, acting as a bridge for downloading the recorded signals and to manage the device during all operational phases (memory cancellation, device status check, device calibration, fixing on the subject, and start or stop recording).

**Figure 3 sensors-22-05031-f003:**
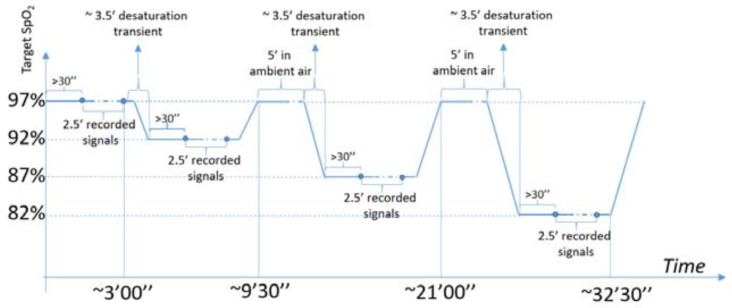
Diagram of the study protocol.

**Figure 4 sensors-22-05031-f004:**
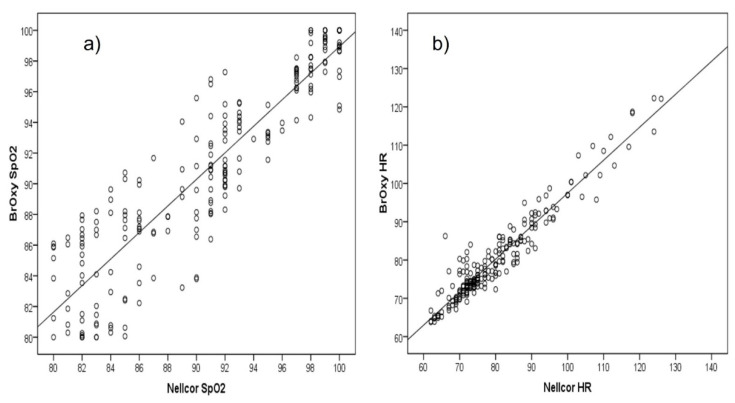
Correlation plot of (**a**) SpO_2_(%) and (**b**) heart rate (HR, bpm) as measured with BrOxy M and Nellcor^TM^.

**Figure 5 sensors-22-05031-f005:**
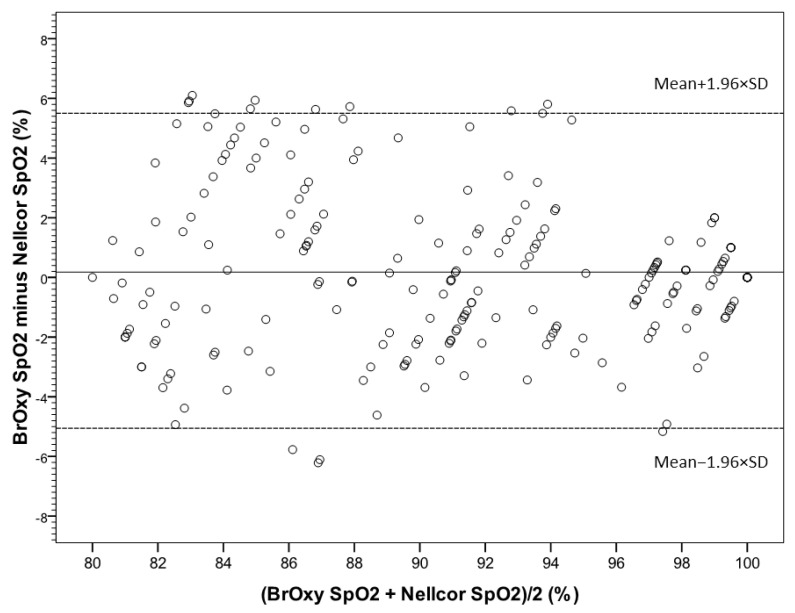
Bland-Altman plot, showing the average values of simultaneous SpO_2_ as measured with BrOxy M and Nellcor^TM^ in *X*-axis versus their differences in *Y*-axis. The mean difference (solid reference line) represents the mean error (bias), and the dashed reference linesare the upper and the lower limits of agreement between BrOxy M and Nellcor^TM^ (−5.1 to 5.5%).

**Figure 6 sensors-22-05031-f006:**
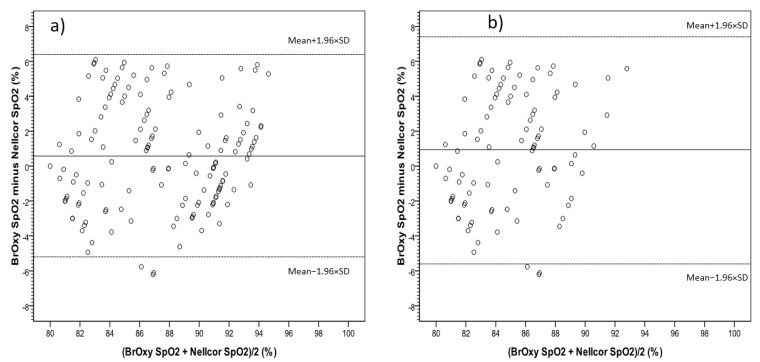
Data pair with Nellcor SpO_2_ ≤ 94% (**a**) and ≤90% (**b**). Bland-Altman plot, showing the average values of simultaneous SpO_2_ as measured with BrOxy M and Nellcor^TM^ in *X*-axis versus their differences in *Y*-axis. The mean difference (solid reference line) represents the mean error (bias), and the dashed reference linesare the upper and the lower limits of agreement between BrOxy M and Nellcor^TM^ (−5.2 to 6.4% for **a**; −5.5 to 7.4% for **b**).

**Figure 7 sensors-22-05031-f007:**
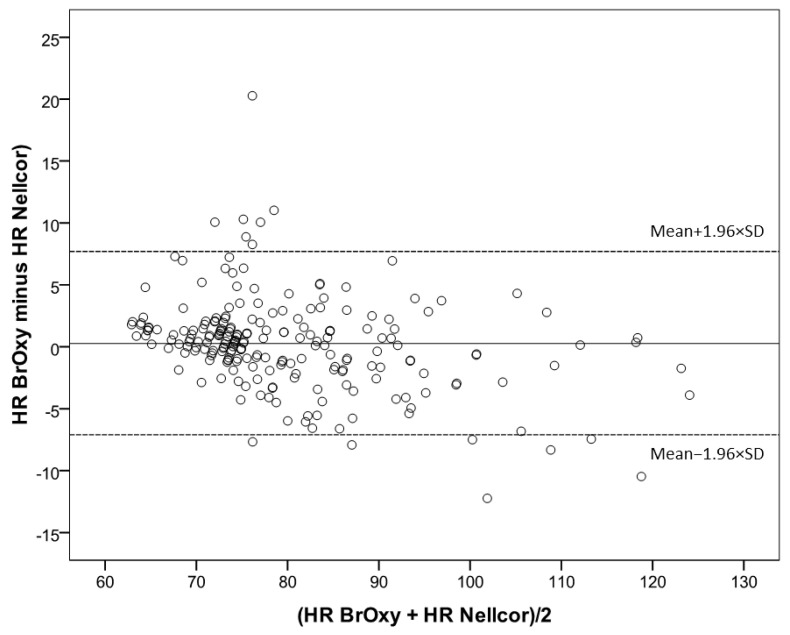
Bland-Altman plot showing the average values of simultaneous Heart rate (HR, bpm) as measured with BrOxy M and Nellcor^TM^ in *X*-axis versus their differences in *Y*-axis. The mean difference (solid reference line) represents the mean error (bias), and the dashed reference linesare the upper and the lower limits of agreement between BrOxy M and Nellcor^TM^ (−7.1 to 7.7 bpm).

**Table 1 sensors-22-05031-t001:** Study protocol.

Plateu n.	Range SpO_2_ (%)	Inhaled Solution	Plateu Duration	N. of Measures of SpO_2_ and HR Extracted from Recordings
(I)	95–100 (target 97%)	Ambient air (medical air)	2.5-’	8
(II)	90–94 (target 92%)	O2 15% + N 85%	2.5-’	8
(III)	85–89 (target 87%)	O2 13% + N 87%	2.5-’	8
(IV)	80–84 (target 82%)	O2 11% + N 89%	2.5’	8
			**Total**	**32 measures**

**Table 2 sensors-22-05031-t002:** Sample characteristics.

	N (%)
* **Participants** *	12
Men	10 (83.3)
Age, years, *mean ± SD [range]*	37 ± 9 (20–51)
Skin color:	
*white*	8 (66.7)
*black (Fitzpatrick scale, type V-VI)*	4 (33.3)
BMI, kg/m^2^, *mean ± SD*	26.2 ± 3.3
* **Data pairs** *	219
Men	183 (83.6)
Age, years, *mean ± SD [range]*	37 ± 9 [20–51]
Skin color:	
*white*	158 (72.1)
*black*	61 (27.9)
BMI, kg/m^2^, *mean ± SD*	26.1 ± 3.3
BrOxy M SpO_2_, %, *mean ± SD*	91.0 ± 6.1
Nellcor SpO_2_%, *mean ± SD*	90.8 ± 6.3
Nellcor SpO_2_ ≤ 94%	147 (67.1)
Nellcor SpO_2_ ≤ 90%	95 (43.4)
BrOxy M HR, bpm, *median [range]*	77 (64–122)
Nellcor HR, bpm, *median [range]*	76 (62–126)

BMI, Body Mass Index; HR, heart rate.

**Table 3 sensors-22-05031-t003:** SpO_2_. Mean error (bias) and accuracy (A_RMS_) of the BrOxy M, in the whole sample and in subjects with defined desaturation.

Nellcor SpO_2_	Bias (95% CI) ^1^ %	A_RMS_ %	Lower Limit of Agreement ^2^ %	Upper Limit of Agreement ^3^ %
80% to 100%	0.18 (−0.17, 0.54)	2.7	−5.1	5.5
≤*94%*	0.57 (0.08, 1.06)	3.0	−5.2	6.4
≤*90%*	0.94 (0.27, 1.61)	3.3	−5.5	7.4

^1^ Mean difference between BrOxy M SpO_2_ and Nellcor SpO_2_; ^2^ mean – SD × 1.96; ^3^ mean + SD × 1.96.

**Table 4 sensors-22-05031-t004:** Heart rate (bpm). Mean error (bias), accuracy (A_RMS_), and mean absolute percent error (MAPE) of the BrOxy M, in the whole sample and in subjects with defined desaturation. Data are reported in percent.

Nellcor SpO_2_	Bias (95% CI) ^1^	A_RMS_	LLA ^2^	ULA ^3^	MAPE ± SD %
80% to 100%	0.25 (−0.24, 0.75)	3.7	−7.1	7.7	3.20 ± 3.3
≤*94%*	−0.01 (−0.65, 0.63)	3.9	−5.2	6.4	3.16 ± 3.4
≤*90%*	−0.39 (−1.12, 0.34)	3.6	−5.5	7.4	3.02 ± 2.7

^1^ Mean difference between BrOxy MSpO_2_ and Nellcor SpO_2_; ^2^ Lower limit of agreement (mean – SD × 1.96); ^3^ Upper limit of agreement (mean + SD × 1.96).

**Table 5 sensors-22-05031-t005:** Multiple Linear regression analysis with difference (BrOxy M SpO_2_ minus Nellcor SpO_2_) as dependent variable.

	r	r^2^	B	95% CI	*p*
SpO2 80% to 100%	0.20	0.041			
*average 1*			0.05	–0.11, 0.01	ns
*age*			0.01	–0.07, 0.05	ns
*sex(ref females)*			0.88	–0.21, 1.97	ns
*skin color(ref black)*			0.16	–1.26, 0.93	ns
*BMI*			0.11	–0.30, 0.08	ns
SpO2 *≤ 94%*	0.23	0.052			
*average 1*			0.06	–0.07, 0.18	ns
*age*			0.02	–0.10, 0.06	ns
*sex(ref females)*			1.31	–0.26, 2.89	ns
*skin color(ref black)*			0.61	–2.26, 1.03	ns
*BMI*			0.14	–0.40, 1.12	ns
SpO2 *≤ 90%*	0.32	0.10			
*average 1*			0.17	–0.08, 0.42	ns
*age*			0.04	–0.08, 0.15	ns
*sex(ref females)*			1.66	–0.80, 4.11	ns
*skin color(ref black)*			0.13	–2.76, 2.50	ns
*BMI*			0.34	–0.76, 0.08	ns

^1^ (BrOxy M SpO_2_ + Nellcor SpO_2_)/2; BMI, Body Mass Index; ns, not significant.

## Data Availability

The data set used in this study is available from the authors upon reasonable request addressed to M.S.

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
