# Peer review of "Accuracy of a New Pulse Oximetry in Detection of Arterial Oxygen Saturation and Heart Rate Measurements: The SOMBRERO Study"

_sensors, 2022, doi:10.3390/s22135031_

Round 1
Reviewer 1 Report
The authors describe a wrist-worn pulse oximeter and its validation in 12 healthy subjects against a fingertip reference device.
However, certain methods used are questionable and the state-of-the art of watch-based oximeters was not properly reviewed.
The current manuscript contains certain sections that can be shortened.
Major issues:
- The authors state that currently no device for around-the-clock SpO2/HR monitoring is available which is wrong. They themselves mention Withings' ScanWatch in the discussion section. In addition they do not mention Oxitone's Oxitone 1000M, the "World’s First FDA-cleared Wrist-Sensor Pulse Oximetry Monitor" (https://www.oxitone.com/oxitone-1000m/). The authors should provide a more complete overview of existing devices and reveal differences of their device to these third-party devices.
- Sensor placement is "preferably on the portion of skin that covers a visible blood vessel".
How is this placement ensured in real-life scenarios, e.g., when an elderly person is using the device? Is there a technical aid for this placement? Could one expect better results from the present study when compared to real-life recordings as the placement was done by a professional?
- There is a double-sided adhesive film placed between the watch and the skin. The authors should state why this is necessary and whether they expect any issues related to this adhesive (skin irritations, itching, redness, etc.) and for how long this adhesive can be applied before changing is necessary. It is questionable to me whether this adhesive is not inferring with the user-friendliness of the envisaged solution. Please comment on that, in particular considering that other products (e.g., Oxitone or Withings ScanWatch) do not need such an adhesive.
- Algorithm processing (lines 187 etc.): could the authors please quantify the thresholds for acceleration, bandpass filtering cutoff frequencies and the calibration parameters (a, b, c) mentioned in this section.
- The calibration procedure mentioned on pages 6 and 7 is expecting a difference between SpO2 measured at the finger vs the wrist. It is difficult to understand the physiological reasoning for such a difference. Please detail the necessity for such a difference and comment on whether this procedure was applied in the current study as this is not obvious to me when reading section 2.5. To me it is wrong to expect a difference in SpO2 at these different measurement sites.
- How often is such a calibration procedure required? Does this mean that for ambulatory measurements the user need to have a fingertip pulse oximeter in addition to the wrist device for regular recalibration?
- Section 2.4 definition of valid measurement: please quantify motion thresholds, band-pass filtering cutoff frequencies, expected range of RED_DC, IR_DC, IR_AC, IR_DC, "certain threshold" of A_{XYZ[t]", etc.
It is further highly questionable to put thresholds on DC or AC values themselves as they highly depend on the amplification gain, LED current, etc. Please also comment on that.
- In my opinion, lines 326 to 336 must go into the results section.
- In my opinion, lines 555 to 576 are not crucial for this publication and are more of introductory nature.
- Pulse oximetry at the wrist is known to suffer from some issues related to positional changes, see 10.1109/EMBC46164.2021.9630185; 10.1109/JPROC.2022.3149785; 10.1016/C2020-0-00098-8. The authors should comment on how these issues are addressed by their device (if at all) and how this has been validated in their protocol (if at all).
Minor issues:
- "pulse rate" (page 3, line 88) is used parallel to "heart rate", please use uniform terminology.
- Please provide number of protocol approved by ethics committee (page 4, section 2.1)
- Please specify the duration of one single measurement: line 312 states 2.5 minutes for 8 measurements so was it 18.75 seconds per measurement?
- Line 325/326: data pairs with error average > 2 * standard deviation were removed, please provide justification/reference for this.
- Line 343: please use Fitzpatrick scale to quantify the pigmentation/skin color of all subjects enrolled to be more precise instead of simply stating "two darkly pigmented subjects". This information must also be added to Table 2.
- Bland-Altman: Lines 374 to ~380: this is common knowledge. Please remove and cite appropriate literature.
- Line 437: please add justification/reference why pulse oximeters should perform better at higher saturation levels.
Author Response
The authors describe a wrist-worn pulse oximeter and its validation in 12 healthy subjects against a fingertip reference device.
However, certain methods used are questionable and the state-of-the art of watch-based oximeters was not properly reviewed.
The current manuscript contains certain sections that can be shortened.
A: We thank the Reviewer for the comments and the useful suggestions in dealing with which we believe to have improved our manuscript.
Major issues:
- The authors state that currently no device for around-the-clock SpO2/HR monitoring is available which is wrong. They themselves mention Withings' ScanWatch in the discussion section. In addition they do not mention Oxitone's Oxitone 1000M, the "World’s First FDA-cleared Wrist-Sensor Pulse Oximetry Monitor" (https://www.oxitone.com/oxitone-1000m/). The authors should provide a more complete overview of existing devices and reveal differences of their device to these third-party devices.
A: The Reviewer is right. However, there is difference between common smartwatches, that are suitable for personal uses (e.g fitness…) and a medical device certified for clinical uses (patients’ monitoring, follow-up, early-diagnosis, and professional healthcare treatments). Our pulse oximeter can be classified as a non-invasive holter-like clinical device for cardio-respiratory patients, and it has to be compared only with other medical certified devices. The usual devices (measuring SpO2.by finger transmission) are invasive and wire, and used only for nocturnal recordings or for the 6MWT (6-minute Walking Test). To our knowledge, among the wrist-wereable devices, there are only a few medical-certified devices (Oxytone 1000; Biobeat medical smartmonitoring; Cardiasense device; Chronisense Medical device; Spry Loop). The continuous monitoring in real daily life can give important information for management/diagnosis of cardio-respiratory diseases. Our BrOxy M aims to be the best solution, with a high-accuracy level of measurements, due to a unique calibration system “tailored on patients”. The references related to the existing devices have been now added in the revised version of the text, as well as the differences between existing devices and the BrOxy M.
- Sensor placement is "preferably on the portion of skin that covers a visible blood vessel".
How is this placement ensured in real-life scenarios, e.g., when an elderly person is using the device? Is there a technical aid for this placement? Could one expect better results from the present study when compared to real-life recordings as the placement was done by a professional?
A: Our BrOxy M is designed to be a medical certified device: it will be placed exclusively by a healthcare professional (e.g. clinician, trained nurse,..), and not directly by the patient. The software device provides a visual wizard which guides the healthcare professional retuning clinical values in real time to best device placement on patient. These procedures are designed in order to guarantee the best quality in clinical value recordings.
- There is a double-sided adhesive film placed between the watch and the skin. The authors should state why this is necessary and whether they expect any issues related to this adhesive (skin irritations, itching, redness, etc.) and for how long this adhesive can be applied before changing is necessary. It is questionable to me whether this adhesive is not inferring with the user-friendliness of the envisaged solution. Please comment on that, in particular considering that other products (e.g., Oxitone or Withings ScanWatch) do not need such an adhesive.
A: Recording clinical values on the wrist - due to the particular physical-physiological conformation of the wrist and the greater mobility of the seat – requires as much stability as possible of the device (all wrist-mounted devices have this problem). Other devices discard or stop recording if a motion is detected, thus producing partial or incomplete monitoring. Even small movements of the device can affect the quality of the recordings. A double-sided adhesive is useful to avoid the movement of the device during the long 24h-monitoring in real-life. The adhesives used are made of biocompatible and disposable material (one adhesive, one test), and they accord to ISO 10993-1 regarding cytotoxicity, allergic sensitization, and skin irritation.
- Algorithm processing (lines 187 etc.): could the authors please quantify the thresholds for acceleration, bandpass filtering cutoff frequencies and the calibration parameters (a, b, c) mentioned in this section.
A: The threshold applied for the identification of the movement condition has been reported in the revised text. We specified that the threshold is expressed in arbitrary units, and it is applied on the absolute value of the instantaneous jolt calculated from the accelerometric signals. We have also reported additional details, such as the type, order, bandwidth of the filters, and the calibration parameters mentioned in this section.
- The calibration procedure mentioned on pages 6 and 7 is expecting a difference between SpO2 measured at the finger vs the wrist. It is difficult to understand the physiological reasoning for such a difference. Please detail the necessity for such a difference and comment on whether this procedure was applied in the current study as this is not obvious to me when reading section 2.5. To me it is wrong to expect a difference in SpO2 at these different measurement sites.
A: Our calibration procedure is designed to “tailor” measurements on specific-unique physical-physiological characteristics (age, dimensions, skin pigmentation,…) of patients submitted to examination. Target of this calibration procedure is to obtain in each case of study the best accuracy in clinical values recording and monitoring. Recording values on wrist is more difficult (see previous observation) that on a finger or any body part subjected to less movements. Also technical approach is different: in finger tips is used a light (red-infrared) transmission way, on wrist is used a light reflection approach. Adopting a calibration technique (once in test session, una-tantum, during the initial placement phase of device on patient) on each single subject allows to calculate a specific value – offset - between finger and wrist measurement. Offset is different and specific for each person and it’s calculated on each patient and allow a dramatical improvement of quality of recording, assuring continuity in monitoring also in critical condition of use). In order to provide the required details, we have modified the section 2.5.
- How often is such a calibration procedure required? Does this mean that for ambulatory measurements the user need to have a fingertip pulse oximeter in addition to the wrist device for regular recalibration?
A: Once in test session, una-tantum, during the initial placement phase of device on patient. No additional fingertip is required: patient during initial device placement phase will put his finger directly on our device sensor windows for some second (90 seconds) in order to detect and calculate the offset parameter.
- Section 2.4 definition of valid measurement: please quantify motion thresholds, band-pass filtering cutoff frequencies, expected range of RED_DC, IR_DC, IR_AC, IR_DC, "certain threshold" of A_{XYZ[t]", etc.
It is further highly questionable to put thresholds on DC or AC values themselves as they highly depend on the amplification gain, LED current, etc. Please also comment on that.
A: It is specified that since the thresholds are expressed in arbitrary units and since they are derived empirically for successive approximations, it does not seem relevant to us to report their value in the tex.
- In my opinion, lines 326 to 336 must go into the results section.
A: Following the suggestion of the Reviewer, we have moved these lines in the results section.
- In my opinion, lines 555 to 576 are not crucial for this publication and are more of introductory nature.
A: the Reviewer is right. We have erased these lines, and partially moved them in the introduction section.
- Pulse oximetry at the wrist is known to suffer from some issues related to positional changes, see 10.1109/EMBC46164.2021.9630185; 10.1109/JPROC.2022.3149785; 10.1016/C2020-0-00098-8. The authors should comment on how these issues are addressed by their device (if at all) and how this has been validated in their protocol (if at all).
A: Positional changes and movements of the patients in real life during monitoring sessions are resolved using a double-sided adhesive between device and wrist. Moreover, the device uses specific algorithms to filtrate and optimize signals and signal features, by correlating the dynamical control with a continuous control of patient body position and its physical activity.
Minor issues:
- "pulse rate" (page 3, line 88) is used parallel to "heart rate", please use uniform terminology.
A: done
- Please provide number of protocol approved by ethics committee (page 4, section 2.1)
A: done
- Please specify the duration of one single measurement: line 312 states 2.5 minutes for 8 measurements so was it 18.75 seconds per measurement?
A: As currently specified in the text, the measurements are instantaneous and are pairs of instantaneous measurements of SpO2 taken at the same time from both BrOxy M and the reference device.
- Line 325/326: data pairs with error average > 2 * standard deviation were removed, please provide justification/reference for this.
A: We have added and reported the reasons that led us to remove the outliers. In particular, we have detailed the test performed after the elimination of the outliers in order to verify the normality of the statistical distribution of the obtained data pairs population.
- Line 343: please use Fitzpatrick scale to quantify the pigmentation/skin color of all subjects enrolled to be more precise instead of simply stating "two darkly pigmented subjects". This information must also be added to Table 2.
A: the requested information about Fitzpatrick scale has been added in the revised text, as well as in Table 2.
- Bland-Altman: Lines 374 to ~380: this is common knowledge. Please remove and cite appropriate literature.
A: following the Reviewer's suggestion, the text describing the Blad&Altman analysis was deleted, and an appropriate reference has been added.
- Line 437: please add justification/reference why pulse oximeters should perform better at higher saturation levels.
A: it is reported in literature that pulse oximeters perform better at the higher saturation levels compared to the lower end (Pretto JJ, Roebuck T, Beckert L, Hamilton G. Clinical use of pulse oximetry: official guidelines from the Thoracic Society of Australia and New Zealand. Respirology. 2014). However, there are no acceptance criteria associated with different levels of hyoxia. Thus, when presenting the ARMS, the common methodology is to provide the data across the whole range (80% to 100%, in the present study). The standard ISO 80601-2-61:2017 (Annex AA) reports that, based on clinical experience, SpO2 accuracy ≤4% is acceptable for many monitoring applications. A specific reference has been added in the revised text.
Reviewer 2 Report
It is an interesting well-written study with valid results. I have no important remarks concerning this paper.
Author Response
A: We thank the Reviewer for this comment.
Reviewer 3 Report
The manuscript is investigating the agreement between a new medical wrist-worn device (BrOxy M) and a reference, medical pulseoximeter (Nellcor PM 1000N). The manuscript suffers from lack of significance and scientific soundness and could be hardly interesting to the readers. I do not recommend it for publication in sensors.
“In addition, it seems that the authors did not put sufficient effort into preparing their manuscript as the paper is under-referenced, conceptually impoverished and poorly written. The English used by the authors differs seriously from normal English, and often produces significant changes in the meaning of sentences. Here, a few places are highlighted to improve the manuscript but suggesting seeking a professional editor before resubmission/publication:
- The agreement between BrOxy M and the reference pulse oximeter was very close 40 to be almost perfect for both SpO2 cut-off equal to 94% (k=0.79) and 90% (k=0.80).
- Currently, no non‐invasive and user-friendly devices are available on the market able to measure continuously and record oxygen saturation and heart rate around the clock.
- “A new device that exploits technology in reflection has been studied and reported in literature which even if plagued by some inconsistencies has shown a good reliability.” Please rewrite the sentence, and use references to the literature mentioned here.
- “Taking into account all these considerations in order to meet the needs of a precise diagnosis and a clear cut continuous monitoring of the above described conditions as well as to fill the gap in available devices so far, our group has designed a pulse oximeter (BrOxy M, CE marked, patented (WO 2019/193196 A1; WO 2021/069729 A1) wearable like a watch and not burdensome for the subject.” This is a very long sentence. Please re-write it.”
Author Response
The manuscript is investigating the agreement between a new medical wrist-worn device (BrOxy M) and a reference, medical pulseoximeter (Nellcor PM 1000N). The manuscript suffers from lack of significance and scientific soundness and could be hardly interesting to the readers. I do not recommend it for publication in sensors.
“In addition, it seems that the authors did not put sufficient effort into preparing their manuscript as the paper is under-referenced, conceptually impoverished and poorly written. The English used by the authors differs seriously from normal English, and often produces significant changes in the meaning of sentences. Here, a few places are highlighted to improve the manuscript but suggesting seeking a professional editor before resubmission/publication:
A: We thank the Reviewer for the comments and the useful suggestions in dealing with which we believe to have improved our manuscript.
- The agreement between BrOxy M and the reference pulse oximeter was very close 40 to be almost perfect for both SpO2 cut-off equal to 94% (k=0.79) and 90% (k=0.80).
A: The sentence has been modified as follows. “The agreement between BrOxy M and the reference pulse oximeter was “substantial” (for SpO2 cut-off 94% and 90%, k=0.79 and k=0.80, respectively)”.
- Currently, no non‐invasive and user-friendly devices are available on the market able to measure continuously and record oxygen saturation and heart rate around the clock.
A: The sentence has been modified as follows “In the last years, many devices have been produced to assist people in their leisure activities. So-called smartwatches are able to measure heart frequency and correlate it with movements. A smaller number of devices has been marketed with improved functions, e.g. measuring not only HR but also SpO2”.
- “A new device that exploits technology in reflection has been studied and reported in literature which even if plagued by some inconsistencies has shown a good reliability.” Please rewrite the sentence, and use references to the literature mentioned here.
A: The sentence has been modified, as folllows “Based on our knowledge, there is at least one study which reported the ability of reflective photoplethysmography in measuring SpO2 levels with adequate accuracy [38], but on our clinical experience, measuring only SpO2 is not useful enough in evaluating patient's situation”
- “Taking into account all these considerations in order to meet the needs of a precise diagnosis and a clear cut continuous monitoring of the above described conditions as well as to fill the gap in available devices so far, our group has designed a pulse oximeter (BrOxy M, CE marked, patented (WO 2019/193196 A1; WO 2021/069729 A1) wearable like a watch and not burdensome for the subject.” This is a very long sentence. Please re-write it.”
A: The sentence has been rewrited as follows: “For this purpose, a new pulse oximeter was designed, wearable like a watch and able to correlate pulse and oximetry with exercise. This new device, named BrOxy M (CE marked, patented: WO 2019/193196 A1; WO 2021/069729 A1), aimed to fill the gap of currently available devices, with a high-accuracy level of measurements, due to a unique calibration system “tailored on patients””.
Reviewer 4 Report
The authors provide accuracy of a new pulse oximetry in detection of arterial oxygen saturation and heart rate measurements. They investigated the agreement between a new medical wrist-worn device (BrOxy M) and a reference medical pulseoximeter (Nellcor PM 1000N). As a wearable device, Broxy M can measure continually SpO2 and HR in everyday life, helping in detecting and following up CVD and CRD subjects. It can be accepted as it is.
Author Response
The authors provide accuracy of a new pulse oximetry in detection of arterial oxygen saturation and heart rate measurements. They investigated the agreement between a new medical wrist-worn device (BrOxy M) and a reference medical pulseoximeter (Nellcor PM 1000N). As a wearable device, Broxy M can measure continually SpO2 and HR in everyday life, helping in detecting and following up CVD and CRD subjects. It can be accepted as it is.
A: We thank the Reviewer for the comments.
Round 2
Reviewer 3 Report
The authors have address all the comments raised by the Reviewer. The manuscript can be accepted for publication, after careful English proof-reading.